# The Importance of Artificial Intelligence in Upper Gastrointestinal Endoscopy

**DOI:** 10.3390/diagnostics13182862

**Published:** 2023-09-05

**Authors:** Dusan Popovic, Tijana Glisic, Tomica Milosavljevic, Natasa Panic, Marija Marjanovic-Haljilji, Dragana Mijac, Milica Stojkovic Lalosevic, Jelena Nestorov, Sanja Dragasevic, Predrag Savic, Branka Filipovic

**Affiliations:** 1Faculty of Medicine Belgrade, University of Belgrade, 11000 Belgrade, Serbia; tijana.glisic78@gmail.com (T.G.); draganamijac@gmail.com (D.M.); drmilicastojkovic@gmail.com (M.S.L.); jelenamartinov@yahoo.com (J.N.); dragasevicsanja@gmail.com (S.D.); psavic-md@hotmail.com (P.S.); branka.filipovic3@gmail.com (B.F.); 2Department of Gastroenterology, Clinical Hospital Center “Dr Dragisa Misovic-Dedinje”, 11000 Belgrade, Serbia; filipovicn@live.com (N.P.); maja-s-92@hotmail.com (M.M.-H.); 3Clinic for Gastroenterohepatology, University Clinical Centre of Serbia, 11000 Belgrade, Serbia; 4General Hospital “Euromedic”, 11000 Belgrade, Serbia; tommilos@hotmail.com; 5Clinic for Surgery, Clinical Hospital Center “Dr Dragisa Misovic-Dedinje”, 11000 Belgrade, Serbia

**Keywords:** artificial intelligence, upper gastrointestinal endoscopy, Barrett’s esophagus, esophageal squam cell carcinoma, gastric cancer, *H. pylori*

## Abstract

Recently, there has been a growing interest in the application of artificial intelligence (AI) in medicine, especially in specialties where visualization methods are applied. AI is defined as a computer’s ability to achieve human cognitive performance, which is accomplished through enabling computer “learning”. This can be conducted in two ways, as machine learning and deep learning. Deep learning is a complex learning system involving the application of artificial neural networks, whose algorithms imitate the human form of learning. Upper gastrointestinal endoscopy allows examination of the esophagus, stomach and duodenum. In addition to the quality of endoscopic equipment and patient preparation, the performance of upper endoscopy depends on the experience and knowledge of the endoscopist. The application of artificial intelligence in endoscopy refers to computer-aided detection and the more complex computer-aided diagnosis. The application of AI in upper endoscopy is aimed at improving the detection of premalignant and malignant lesions, with special attention on the early detection of dysplasia in Barrett’s esophagus, the early detection of esophageal and stomach cancer and the detection of *H. pylori* infection. Artificial intelligence reduces the workload of endoscopists, is not influenced by human factors and increases the diagnostic accuracy and quality of endoscopic methods.

## 1. Introduction

In recent years, there has been a growing interest in the application of artificial intelligence (AI) in medicine, especially in specialties where visualization methods are applied. The most significant development of AI is taking place in the fields of radiology, gastroenterology (endoscopy), surgery and dermatology, but also in other specialties. The beginning of AI dates back to 1950 [1].

Artificial intelligence is defined as the ability of a computer to achieve human cognitive performance, primarily learning and decision making [2,3]. To achieve this, it is necessary to enable computers to “learn”. There are two ways that allow computers to perform specific operations. One is classic programming, where based on predefined algorithms (programs), the computer determines output data based on input data. Another much more complex approach is machine learning (ML), and it is the basis of AI.

In classic machine learning, during programming, mathematical descriptions of patterns (e.g., color, texture, edge, size, etc.) are defined, and the computer further classifies existing data (hand-crafted algorithm) [4,5]. The program itself is insufficient to enable the differentiation of output variables solely based on input data. In order to achieve further ML, “training” is necessary, attained by processing a large number of different input data, which the computer “learns”. In endoscopy, these inputs are images or videos. After the training phase, the computer is able to recognize certain features even in images or videos that are unknown to it. There are different ML models, which can be unsupervised and supervised [3]. The following concepts can be used: support vector machines (SVMs), decision trees and artificial neural networks [3].

A more complex machine learning system is deep learning (DL). The most commonly used system for DL is the convolutional neural network (CNN) [5]. The schematic structure of the CNN system is shown in Figure 1.

This system mimics the human neural network. It consists of a large number of artificial neurons, which are organized into an artificial neural network. Namely, neurons are classified into layers and the so-called multilayered system. The neurons of one layer are connected to the neurons of the next layer and the output data of each neuron have the function of input data for the neurons of the next layer [4,6]. During DL, the computer itself extracts the data, thus forming its recognition patterns, without the influence of the programmer [5]. However, the disadvantage of the deep learning system is that it remains unknown how the computer makes individual decisions (black box) [5]. Unlike conventional ML, which requires human intervention to correct errors, the DL system has the ability to learn from its errors [2].

The application of artificial intelligence in endoscopy refers to computer-aided detection (CAD) and the somewhat more complex computer-aided diagnosis (CADx).

The development of AI methods in upper gastrointestinal endoscopy is focused in three segments [7]:Quality assessment;Detection of lesions;Characterizations of lesions.

These three segments follow the endoscopist’s cognitive process. Namely, first it is necessary to perform a quality examination, followed by the detection of lesions and their characterization. By integrating AI algorithms, the diagnostic process is significantly improved. Quality assessment refers to the adequate visualization of all anatomical landmarks, with the assistance of AI methods (e.g., multi-frame classification) [7]. The application of AI in upper endoscopy is primarily aimed at improving the detection of premalignant and malignant lesions. This is especially important if it is known that during endoscopy a significant part of cancer can be undiagnosed. Namely, the frequency of missed malignancies was evaluated in a study that included 4,105,399 patients [8]. Carcinomas diagnosed <6 months after upper endoscopy are marked as prevalent, and those diagnosed up to 6 to 36 months are marked as missed [8]. The highest percentage of missed esophageal cancers was for adenocarcinoma (6.1%), while for squamous cell carcinoma it was 4.2% [8]. The majority of missed gastric cancers were adenocarcinomas, at 5.7% [8]. Similar results were obtained in some other studies [9,10].

Current research in the field of AI application in upper gastrointestinal endoscopy is focused on the detection, demarcation and characterization of esophageal and stomach cancer, including premalignant conditions (Barrett’s esophagus, *H. pylori* infection, etc.) [2]. The emphasis is placed on early diagnosis of these diseases.

## 2. Barrett’s Esophagus and Esophageal Adenocarcinoma

Barrett’s esophagus (BE) represents the replacement of squamous epithelium of the esophagus by metaplastic columnar epithelium [11]. Since metaplasia is present in Barrett’s esophagus, this is a premalignant condition and can lead to esophageal adenocarcinoma (EAC). The progression of BE to EAC is below 1% per patient year [2,12,13].

During endoscopic exploration of the esophagus, it is necessary to determine the level of the esophagogastric junction and the Z line. Under normal conditions, the esophagogastric junction and the Z line are at the same level. If there is an extension of the cylindrical epithelium by more than 1 cm from the proximal end of the gastric folds, BE is suspected [14]. The diagnosis is confirmed by the histopathological finding of specialized intestinal metaplasia [14]. When taking biopsies of suspected Barrett’s esophagus, the Seattle protocol is applied. Namely, in patients with non-dysplastic BE, it is recommended to take biopsies from four quadrants of the esophagus, every 2 cm, starting from the esophagogastric junction [15]. In patients with BE and low-grade dysplasia, biopsies of all four quadrants, every 1–2 cm, starting from the esophagogastric junction are recommended, while in the case of BE with high-grade dysplasia, biopsies of all four quadrants, every 1 cm, from the esophagogastric junction are recommended [15]. Biopsies of all observed changes (nodules, depressions, irregularities, etc.) are mandatory. Barrett’s esophagus is further classified based on the Prague classification [16]. For its application, it is necessary to determine the level of the esophagogastric junction, the circumferential Barrett’s esophagus and the maximum extension (tongue) of the Barrett’s esophagus. The difference (in centimeters) between the esophagogastric junction and the circumferential (C), i.e., tongue of Barrett’s esophagus (M) is classified as C (centimeters) M (centimeters).

In order to make a correct diagnosis of BE, it is necessary to conduct a careful and detailed examination of the esophagus. The method of choice is high-definition white-light endoscopy (HD-WLE), but classical (dye) and virtual chromoendoscopy methods also play a significant role [12,17]. From dye chromoendoscopy, the use of 1.5–3.0% acetic acid is useful, because dysplastic tissue has an accelerated loss of aceto-whitening, and it enables easier differentiation from normal tissue [18]. In addition, better visualization is enabled by virtual chromoendoscopy techniques: NBI (narrow-band imaging), BLI (blue laser imaging), LCI (linked color imaging) and others [17].

Given that a detailed examination is necessary and discrete mucosal changes need to be observed, significant assistance could be provided by AI. Namely, by applying AI techniques, the detection, diagnosis and endoscopic treatment of BE may be improved [6]. The importance of early detection of dysplasia and carcinoma in BE is in the outcome and different therapeutic modalities. Specifically, advanced EAC has a poor prognosis and requires invasive treatment (most often surgical treatment), while earlier stages of the disease (stage T1) allow endoscopic resection [13,19,20].

The application of AI in the diagnosis of BE is aimed at the detection of lesions, their characterization and the assessment of the depth of invasion (if cancer is present) [6].

The first study on the application of AI in the detection of early BE neoplasia was published by van der Sommen et al. in 2016 [21]. In their research, 100 high-definition (HD) endoscopic images (44 patients) were used and further analyzed by a total of five expert endoscopists. One endoscopist had access to the results of the histopathological analysis (nonblinded) and his findings were the “gold standard”, while the others did not know the pathohistological findings [21]. The computer system was constructed using the SVM model, based on the color and texture of the region of interest [21]. The mentioned system showed a sensitivity and specificity of 86% and 87%, respectively [21]. De Groof et al. developed and validated a computer-aided detection system for early neoplasia in BE [12]. The authors used a CAD system based on the ResNet/UNet hybrid model. The ResNet model is used for image classification, while the UNet model is used for intra-image prediction segmentation [12]. The study indicated that the mentioned CAD system enables the classification of endoscopic images into non-dysplastic and dysplastic BE with an accuracy of 89%, a sensitivity of 90% and a specificity of 88% [12]. In addition to the detection of neoplasia, this system allows determining the optimal localization of the biopsy site in 97% of cases [12]. This is very significant because the histopathological analysis of the altered region is crucial for diagnosis establishment. The application of AI in BE neoplastic detection and delineation shows performance similar to that of expert endoscopists, which is higher than that of non-expert endoscopists [12,22]. Fockens et al. also proved that the CAD system with high sensitivity (depending on the datasets, 88% and 100%) has better performance compared to general endoscopists, but with slightly lower specificity (64–66%) [23].

Most of the developed AI models for the detection of neoplastic BE are image-based, that is, they are based on analyzing static images. With the further development of AI, systems enabling the analysis of real-time video sequences were formed [9,10,24]. These systems are much closer to everyday clinical practice. Abdelrahmin et al. have developed and validated a CAD system that enables the real-time detection of neoplastic BE with an accuracy of 92.0%, a sensitivity of 93.8% and a specificity 90.7% [24]. Furthermore, the CAD system showed significantly better accuracy, sensitivity and specificity compared to endoscopists [24].

Narrow-band imaging, as a method of virtual chromoendoscopy, enables a clearer visualization of the mucosal and vascular pattern. The application of this technique improves the visualization of early neoplastic lesions compared to classic WLE (white-light endoscopy), especially if it is used in combination with magnification (zoom endoscopy). Struyvenberg et al. conducted a study in which they evaluated the performance of the CAD system with the application of NBI zoom endoscopy in the detection of neoplastic BE [25]. The results of this study showed that the application of a video-based CAD system, along with the technique of NBI zoom endoscopy, had an accuracy of 83%, a sensitivity of 85% and a specificity of 83% [25]. As the main limitation of the study, the authors mentioned the small number of NBI zoom images that are available for “learning” the CAD system, which is understandable when it is known that the majority of datasets are images obtained by WLE. Swagner et al. developed an algorithm for the CAD of early BE neoplasia using volumetric laser endomicroscopy [26]. This algorithm showed good performance, and its importance as assistance to the endoscopists in their clinical work was pointed out [26].

Lui et al. conducted a meta-analysis in which the pooled sensitivity of AI techniques in the diagnosis of neoplastic BE was 88.0% (95% CI, 82.0–92.1%), the specificity was 90.4% (95% CI, 85.6–94.5%) and the area under the curve (AUC) was 0.96 (95% CI, 0.93–0.99) [22]. Additionally, there were no significant differences in the different modalities of endoscopy (WLE vs. volumetric laser endomicroscopy), nor in AI methods (CNN vs. non-CNN) [22]. Similar results were obtained in the meta-analysis conducted by Vissagi et al. [20]. These findings concluded that AI methods in the diagnosis of Barrett’s neoplasia have a sensitivity of 89%, a specificity of 86%, a positive likelihood ratio (PLR) of 6.50, a negative likelihood ratio (NLR) of 0.13, a diagnostic odds ratio (DOR) of 50.53 and an area under the summary receiver operating characteristic curve (AUROC) of 90% [27]. In the aforementioned study, no significant difference in performance between AI and endoscopists was recorded if WLE methods were used [27].

A summary of the results of selected studies in the application of AI for the detection of neoplastic BE is shown in Table 1 [9,10,12,21,23,24,25,26,28].

## 3. Esophageal Squamous Cell Carcinoma (ESCC)

Squamous cell carcinoma is the most common esophageal carcinoma [13,29]. Overall 5-year survival in this type of cancer is 15–25%, with better prognosis if the disease is detected at an earlier stage [13]. Also, detecting ESCC in earlier stages allows performing less aggressive treatment modalities (endoscopic therapy) [20]. The method of choice in the diagnosis of ESCC is upper gastrointestinal endoscopy. Dye and virtual chromoendoscopy methods also contribute to early diagnosis [20,30,31]. The application of dye chromoendoscopy with Lugol’s solution is particularly significant, which enables easier detection of esophageal squamous dysplasia [32]. Staining with Lugol’s solution enables the demarcation of the altered mucosa of the esophagus, because the mucosa containing early ESCC is not stained with this solution and due to glycogen depletion [19]. Although this method of dye chromoendoscopy represents the “gold standard” in the diagnosis of early squamous cell neoplasia, with a high sensitivity (over 90%), the specificity of this method is about 70% [5]. The reason for the lower specificity is the fact that some benign diseases can cause glycogen depletion, and therefore cannot be stained with Lugol’s solution. The appearance of dysplastic epithelium in the esophagus is quite difficult to detect with the use of WLE, since macroscopic changes (nodules, plaques and ulcerations) seen in advanced ESCC are generally not present. In order to improve the percentage of detection of esophageal dysplasia and early ESCC, AI techniques are also being developed.

The beginning of the application of computer assistance in the diagnosis of ESCC occurred in 2007, when Kodashima et al. developed a computer analysis system based on endocytoscopy [33]. It enabled easier differentiation of malignant from non-malignant esophageal tissue. However, this method was based on the computer analysis of images of the nuclear region of cells but did not involve AI.

One of the more significant studies in the field of AI application in the early diagnosis of esophageal cancer was published in 2018 by Horia et al. [34]. They used WLE and NBI as the endoscopic methods and a CNN as the deep learning method. The sensitivity of the developed method was 98%, with the detection of all lesions smaller than 10 mm [34]. In the aforementioned study, NBI showed higher sensitivity compared to WLE (89% vs. 81%) [34]. The diagnostic accuracy of the aforementioned AI system in the detection of superficial esophageal cancer was 99%, or 92% for advanced cancer [34]. The authors noted that the CNN could not adequately register some cases of esophageal cancer if the surrounding mucosa was inflamed [34]. In the manuscript by Feng et al., the application of a CNN on images obtained by WLI (white-light imaging) showed a sensitivity of 90.1%, a specificity of 94.3%, an accuracy of 88.3%, a PPV of 88.3% and an NPV of 94.7% [35]. Similar results were obtained by Wang et al. [36]. In their study, Feng et al. used the YOLOv5l model as a deep learning method. Depending on the endoscopy method, it was concluded that NBI has better performance compared to WLE but is inferior to dye chromoendoscopy with Lugol’s solution [35]. Otherwise, the obtained data are comparable to the performance of expert endoscopists, while they are significantly higher than those of less experienced endoscopists (junior and mid-level endoscopists) [35].

In addition to the detection of early ESCC, it is necessary to determine the depth of the invasion. This is necessary in order to determine the adequate therapeutic modality (surgery vs. endoscopy). Shimamoto et al. developed an AI model using a CNN to estimate the depth of ESCC invasion in real time [37]. This is the first study in which data extraction from video images was used. Their method showed a sensitivity of 71%, a specificity of 95% and an accuracy of 89% if ME was used along with WLE [37]. These performances are comparable to or better than expert endoscopists, depending on whether ME is used in addition to WLI or not [37].

Yuan et al. incorporated WLE, ME-NBI and Lugol’s solution staining into their AI model for the early detection of superficial ESCC [38]. In this multicenter study, they concluded that the application of AI enables the detection of this type of cancer with an accuracy of 91.1%, a sensitivity 96.9.7% and a specificity 83.9% for all investigated endoscopic imaging modalities [38]. This is a study that included the most different endoscopic modalities, and in addition to endoscopic images, it also included video analysis.

The pooled performance of the AI method in the detection of neoplastic lesions of the esophageal squamous epithelium is a sensitivity of 75.6% (95% CI, 48.3–92.5%), a specificity of 92.5% (95% CI, 66.8–99.5%) and an AUC of 0.88 (95% CI, 0.82–0.96) [32]. The results from this study favor the use of NBI over WLE [22]. In the meta-analysis by Vissagi et al., data were obtained that the application of AI techniques in the diagnosis of squamous cell carcinoma of the esophagus has a sensitivity of 95%, a specificity of 92%, a PLR of 12.65, an NLR of 0.05, a DOR of 258.36 and an AUROC of 97% [27]. The performance of the AI method is slightly better than the performance of the endoscopist, but without a significant difference [27].

The results of selected studies on the application of AI in the diagnosis of ESCC are shown in Table 2 [34,35,36,37,38,39,40,41].

## 4. Early Gastric Carcinoma

Gastric cancer is the fifth most common cancer worldwide [42]. Five-year survival depends on the stage at which the disease was detected. For advanced gastric cancer, 5-year survival is 5–25%, while for early it is over 90% [43]. Early gastric cancer does not penetrate the gastric wall deeper than the submucosa, regardless of lymph node metastases [44]. There are two types of gastric cancer, intestinal and diffuse [45]. The therapeutic approach and prognosis of these two types of cancer are different.

In the early stages, gastric cancer is usually asymptomatic. In the later evolution of the disease, the following may occur: dyspeptic symptoms, abdominal pain, nausea, vomiting, disgust for meat, weight loss, anemia, bleeding, etc. [45]. The diagnosis is made on the basis of upper gastrointestinal endoscopy and a pathohistological analysis of gastric biopsies. Endoscopic presentation of early gastric cancer can be in the form of red discoloration of part of the gastric mucosa, gastric ulceration or depressed gastric lesion [46]. Since these lesions are often discrete and difficult to visualize, advanced endoscopic methods can help in diagnosis. The methods of dye and virtual chromoendoscopy, as well as the application of ME, have better performance in detecting early gastric cancer compared to classic WLE [46,47,48].

Miyaki et al. published one of the first studies on the application of AI in the detection of early gastric cancer in 2013 [49]. The constructed system was trained on a total of 493 endoscopic images, of which 235 were images without neoplastic tissue and 258 had gastric cancer present. In the training sample with cancer, 67% of the samples were with differentiated cancer and 33% with undifferentiated cancer [49]. This system showed an accuracy of 85.9%, a sensitivity of 84.8%, a specificity of 87.0%, a PPV of 86.7% and an NPV of 85.1% [49].

The detection of early gastric cancer with the assistance of AI can be performed on previously obtained endoscopic images but also in real time. Luo et al. developed and validated the GRAIDS (Gastrointestinal Artificial Intelligence Diagnostic System) for the diagnosis of upper gastrointestinal cancers [50]. In this multicenter study, the authors developed the system using Deep Lab’s V3+ concept. The system was trained and validated on a total of 1,036,496 endoscopic images (84,424 patients), so this is the largest study on the use of AI in the diagnosis of cancer of the upper gastrointestinal tract [50]. The results indicated the excellent performance of this model, with the possibility of real-time use.

Early detection of gastric cancer, in addition to enabling better survival, also enables the application of less invasive but curative methods compared to advanced cancer. The main criterion used when evaluating the possibility of curative endoscopic resection is the depth of invasion. The first study in the application of CAD to assess the depth of gastric cancer invasion based on endoscopic images was published by Kubota et al. [51]. This system showed an overall accuracy of 64.7%, which is slightly higher for the T1 stage (77.2%) and lowest for the T2 stage (49.1) [51].

In a study by Niikura et al., the effectiveness of AI and expert endoscopists in the detection of gastric cancer was compared [52]. In a sample of 500 patients (100 with gastric cancer and 400 without cancer), AI detected cancer in 100% of patients and expert endoscopists in 94.1% of cases [52]. Early cancer was diagnosed in 100% in the AI group and in 88.4% in the expert group, while success in detecting invasive cancer (T1b stage and higher) was 100% in both groups [52]. Although there is a difference in the detection of early cancer, it is not statistically significant.

A meta-analysis that assessed the performance of AI in the detection of neoplastic gastric lesions indicated a pooled sensitivity of these techniques of 92.1% (95% CI, 87.7–95.4%) and a specificity of 88.0% (95% CI, 78.0–95.0%) with an AUC of 0.96 (95% CI, 0.94–0.99) [22]. There was no significant difference in the different modalities of endoscopy (WLE vs. NBI) or in the AI method (CNN vs. support vector model) [22].

The excellent performance of AI methods in the diagnosis of gastric cancer can also be explained by the characteristic morphological characteristics of these tumors. Namely, according to the Paris classification, gastric carcinomas are most often type IIa (elevated lesion), alone or in combination with type IIc (depressed lesion), as IIa+IIc or IIc+IIa [53]. Changes in the mucosal and vascular pattern are certainly important.

The application of AI methods, in addition to endoscopy, can be used in pathohistological and CT diagnoses of gastric cancer, surgical treatment and predicting the outcome of this disease [43,54].

The results of selected studies on the application of AI in the diagnosis of early gastric cancer are shown in Table 3 [49,50,55,56,57,58,59].

## 5. *H. pylori* Gastritis

*Helicobacter pylori* is a microaerophilic Gram-negative bacterium. It is estimated that the infection is present in half of the world’s population [60]. *H. pylori* can lead to chronic gastritis, intestinal metaplasia, MALT (mucosa-associated lymphoid tissue) lymphoma and gastric cancer [60]. In patients with *H. pylori*-induced gastritis, endoscopic findings include mucosal edema, diffuse hyperemia, thickening of gastric folds, mucosal nodularity and atrophy [61,62]. The regular arrangement of collecting venules and fundic gland polyps are characteristic of *H. pylori* negative mucosa [62]. However, the endoscopic findings are not specific, and the diagnosis of *H. pylori* infection can be confirmed by a histopathological analysis of gastric biopsies or non-invasive tests (urea breath test, stool antigen test, serology and molecular methods). For the histopathological diagnosis of *H. pylori*-positive gastritis, biopsies are used according to the updated Sidney protocol [15]. It involves taking two biopsies from the antrum, two biopsies from the gastric corpus and one biopsy from the angulus. If biopsies are taken only for the diagnosis of *H. pylori* infection, 1–2 biopsies from the antrum are sufficient [15].

AI systems can improve the optical diagnosis of *H. pylori* infection based on pattern recognition applied to endoscopic images [62]. Further refinement and development of the AI system would help in much faster and more accurate diagnosis, but also in avoiding unnecessary gastric biopsies in order to detect *H. pylori* infection.

Shischijo et al. developed an AI system, which they applied to a total of 397 patients, of which 72% were *H. pylori*-positive [63]. To develop this detection system, they used GooGLeNet, a 22-layer CNN [63]. This system showed an accuracy of 87.7%, a sensitivity of 88.9% and a specificity of 87.4% [63]. By comparing the performance of the AI system and the endoscopist, the authors concluded that the accuracy of the AI system was higher, with a shorter detection time, while the sensitivity and specificity were comparable [63].

An improvement in the performance of AI in the detection of *H. pylori* infection can be achieved if multiple endoscopic images are used, as well as if advanced endoscopy techniques are used. In a pilot study by Zheng et al., it was found that the application of multiple stomach images improves the accuracy (93.8% vs. 84.5%), sensitivity (91.6% vs. 81.4%) and specificity (98.6% vs. 90.1%) of endoscopy compared to the analysis of one image [64].

The use of advanced endoscopy techniques, primarily NBI, BLI, LCI and ME, leads to the improved detection of *H. pylori* infection [60,65]. Nakashima et al. showed that the use of advanced endoscopic methods is superior to classical WLI in the detection of *H. pylori*-positive gastritis [66]. In this study, which was conducted on a sample of 222 patients, it was concluded that the AUC was significantly higher for BLI-bright (0.96) and LCI (0.95) compared to WLI (0.66) [66]. The advantage of LCI endoscopy in the detection of *H. pylori* infection was confirmed by the same author in a subsequent study [67].

Based on the results of the meta-analysis, the pooled sensitivity of the AI method in the detection of *H. pylori* infection was 83.9% (95% CI, 70.8–92.9%), while the specificity was 89.7% (95% CI, 79.4–95.9%) and the AUC was 0.92 (95% CI, 0.88–0.97) [22].

The summary of the results of selected studies in the use of AI for the detection of *H. pylori* infection are shown in Table 4 [63,64,66,67,68,69,70,71].

## 6. Conclusions

Artificial intelligence reduces the workload of endoscopists, is not influenced by human factors (e.g., fatigue, stress, etc.) and contributes to increasing the diagnostic accuracy and quality of endoscopic methods. These systems are very effective in the diagnosis of neoplastic Barrett’s esophagus, esophageal squamous cell carcinoma, early gastric cancer and *H. pylori* infection. AI methods are effective with the use of different endoscopy modalities. Given that the lack of time is a significant enemy of endoscopy, computerized data processing can speed up the detection time, because it processes more data at a higher speed than a human being.

However, these systems also have their weaknesses. Given that AI systems, after development, do not include constant human supervision and correction, purely technical errors are possible. Namely, during the application of AI methods, false-negative and false-positive findings may occur. More important are false-negative results, which most often occur due to visualization and technical errors.

The question often arises whether AI will replace doctors. At this stage of the development of science and technology, it is unlikely. Namely, for an expert endoscopist, in addition to endoscopy knowledge, significant clinical experience is also necessary. This has been proven in studies where the performance of expert endoscopists is comparable to the performance of AI. Also, the opinion of experts was used as the “gold standard” in studies for the application of AI. Further improvement of these techniques will provide significant help in diagnosis and facilitate the learning of endoscopy for all endoscopists, especially less experienced ones. Therefore, endoscopists should not only rely on such systems, but these systems should serve as their assistants.

### Future Directions

Most of the previous research in this area has been conducted using WLE, while studies using advanced endoscopic techniques are sporadic. Therefore, future directions should be focused on the use of advanced endoscopic techniques, primarily virtual chromoendoscopy. Unfortunately, the main limiting factor is the unavailability of this technique and trained endoscopists in a large number of hospitals. For an adequate assessment of the clinical application of AI methods, a larger number of prospective studies are needed, as well as a larger number of studies that include the analysis of real-time images and videos. Also, we believe that it would be interesting to develop models that include certain clinical, laboratory and radiological data.

## Figures and Tables

**Figure 1 diagnostics-13-02862-f001:**
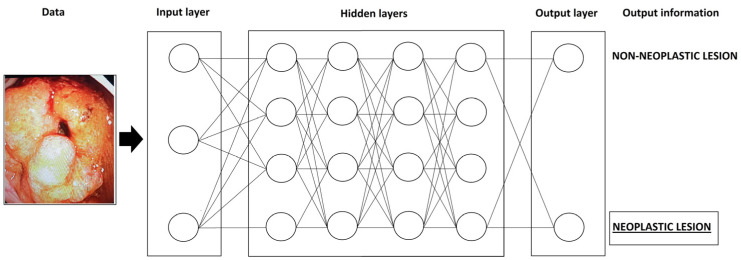
The schematic structure of the CNN system.

**Table 1 diagnostics-13-02862-t001:** Summary of selected studies on the application of artificial intelligence in the diagnosis of neoplastic Barrett’s esophagus.

Authors (Reference)	Year	Country	Design	Dichotomous Variable	Endoscopic Methods	AI Method	Performance		
							Accuracy (%)	Sensitivity (%)	Specificity (%)
van der Sommen [21]	2016	Netherlands	Retrospective	neoplastic/non-neoplastic BE	WLI	SVM	N/A	86.0	87.0
de Groof [12]	2020	Netherlands	Retrospective	neoplastic/non-neoplastic BE	WLI	ResNet/UNet (CNN)	89.0	90.0	88.0
Fockens [23]	2023	Netherlands	Prospective (multicentric)	neoplastic/non-neoplastic BE	WLI	CNN	N/A	100.0	66.0
Abdelrahim [24]	2023	UK	Retrospective (multicentric)	neoplastic/non-neoplastic BE	WLI	CNN	92.0	93.8	90.7
Struyvenberg [25]	2021	Netherlands	Retrospective (multicentric)	neoplastic/non-neoplastic BE	NBI	ResNet/UNet (CNN)	84.0	88.0	78.0
de Groof [9]	2020	Netherlands	Prospective	neoplastic/non-neoplastic BE	WLI	ResNet/UNet (CNN)	90.0	91.0	89.0
Ebigdo [10]	2020	Germany	Prospective	neoplastic/non-neoplastic BE	WLI	CNN	89.9	83.7	100
Hashimoto [28]	2019	USA	Retrospective	neoplastic/non-neoplastic BE	WLI	CNN	N/A	98.6	88.8
					NBI		N/A	92.4	99.2
Swager [26]	2017	Netherlands	Retrospective	neoplastic/non-neoplastic BE	VLE	CNN	N/A		

N/A—not available; AI—artificial intelligence; BE—Barrett’s esophagus; WLI—white-light imaging; SVM—support vector machine; CNN—convolutional neural network; UK—United Kingdom; NBI—narrow-band imaging; USA—United States of America; VLE—volumetric laser endomicroscopy.

**Table 2 diagnostics-13-02862-t002:** Summary of selected studies on the application of artificial intelligence in the diagnosis of esophageal squamous cell carcinoma.

Authors (Reference)	Year	Country	Design	Dichotomous Variable	Endoscopic Methods	AI Method	Performance		
							Accuracy (%)	Sensitivity (%)	Specificity (%)
Horie [34]	2019	Japan	Retrospective	cancer/non-cancer	WLI	CNN	N/A	98.0 *	79.0 **
Feng [35]	2023	China	Retrospective	cancer/non-cancer	WLI	CNN	88.3	90.1	94.3
Wang [36]	2023	China	Retrospective	cancer/non-cancer	WLI	YOLOv5l	96.9	87.9	98.3
					NBI		98.6	89.3	99.5
					LCE		93.0	77.5	98.0
Shimamoto [37]	2020	Japan	Retrospective	depth of invasion	WLI	CNN	87.3	50.0	98.7
					ME		89.2	70.8	94.9
Yuan [38]	2022	China	Retrospective (multicentric)	superficial carcinoma/non-carcinoma	WLI	CNN	86.6	93.3	78.5
					non-ME NBI		91.7	98.0	85.1
					ME-NBI		96.5	99.4	89.0
					Iodine staining		92.2	96.7	86.9
Ohmori [39]	2020	Japan	Retrospective	cancer/non-cancer	WLI	CNN	81.0	90.0	76.0
					NBI/BLI		77.0	100.0	63.0
					ME		77.0	98.0	56.0
Guo [40]	2019	India	Retrospective	cancer/non-cancer	NBI	SegNet	N/A	98.0	95.0
Nakagwa [41]	2019	Japan	Retrospective	cancer/non-cancer	WLI	Single Shot MultiBox	91.0	90.1	95.8

* For each case; ** for each image. N/A—not available; AI—artificial intelligence; WLI—white-light imaging; CNN—convolutional neural network; YOLOv5l—model “You Only Look Once” large extension; NBI—narrow-band imaging; LCE—Lugol chromoendoscopy; ME—magnifying endoscopy, BLI—blue laser imaging.

**Table 3 diagnostics-13-02862-t003:** Summary of selected studies on the application of artificial intelligence in the diagnosis of early gastric carcinoma.

Authors (Reference)	Year	Country	Design	Dichotomous Variable	Endoscopic Methods	AI Method	Performance		
							Accuracy (%)	Sensitivity (%)	Specificity (%)
Miyaki [49]	2013	Japan	Retrospective	cancer/non-cancer	ME-FICE	SVM	85.9	84.8	87.0
Luo [50]	2019	China	Retrospective (multicentric)	cancer/non-cancer	WLI	CNN	92.7	94.6	91.3
Tang [55]	2020	China	Retrospective	cancer/non-cancer	WLI	CNN	85.1–91.2	85.9–95.5	81.7–90.3
Ikenoyama [56]	2021	Japan	Retrospective	cancer/non-cancer	WLI, ICE, NBI	CNN	N/A	58.4	87.3
Nagao [57]	2020	Japan	Retrospective	depth of invasion	WLI	CNN	94.4	84.4	99.3
					NBI		94.3	75.0	100.0
					ICE		95.5	87.5	100.0
Kanasaka [58]	2018	Taiwan	Retrospective	cancer/non-cancer	ME-NBI	SVM	96.3	96.7	95.0
Zhu [59]	2018	USA	Retrospective	depth of invasion	WLI	CNN	89.1	76.4	95.5

N/A—not available; AI—artificial intelligence; ME—magnifying endoscopy; FICE—flexible spectral imaging color enhancement; WLI—white-light imaging; CNN—convolutional neural network; ICE—indigo-carmine chromoendoscopy; SVM—support vector machine; USA—United States of America.

**Table 4 diagnostics-13-02862-t004:** Summary of selected studies on the application of artificial intelligence in the diagnosis of *H. pylori*.

Authors (Reference)	Year	Country	Design	Dichotomous Variable	Endoscopic Methods	AI Method	Performance		
							Accuracy (%)	Sensitivity (%)	Specificity (%)
Shichijo [63]	2017	Japan	Retrospective	Presence/absence *H. pylori*	WLI	CNN	87.7	88.9	87.4
Zheng [64]	2019	China	Retrospective	Presence/absence *H. pylori*	WLI	CNN (Res-Net 50)	93.8 *	91.6 *	98.6 *
Seo [68]	2023	Korea	Retrospective (multicentric)	Presence/absence *H. pylori*	WLI	CNN	94.0 **	96.0 **	90.0 **
							88.0 ***	92.0 ***	79.0 ***
Nakashima [67]	2020	Japan	Prospective	Presence/absence *H. pylori*	WLI	CNN	77.5	60.0	86.2
					LCI		82.5	62.5	92.5
Li [69]	2023	China	Retrospective	Presence/absence *H. pylori*	WLI	CNN	84.0	82.0	86.0
Yasuda [70]	2019	Japan	Retrospective	Presence/absence *H. pylori*	LCI	SVM	87.6	90.5	85.7
Itoh [71]	2019	Japan	Prospective	Presence/absence *H. pylori*	WLI	CNN	N/A	86.7	86.7
Nakashima [66]	2018	Japan	Prospective	Presence/absence *H. pylori*	WLI	CNN	N/A	66.7	60.0
					BLI-bright		N/A	96.7	86.7
					LCI		N/A	96.7	83.3

* For multiple images; ** for Korean; *** for non-Korean. AI—artificial intelligence; WLI—white-light imaging; CNN—convolutional neural network; LCI—linked color imaging; SVM—support vector machine; BLI—blue laser imaging.

## Data Availability

No new data generated.

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
