# Peer review of "The Importance of Artificial Intelligence in Upper Gastrointestinal Endoscopy"

_diagnostics, 2023, doi:10.3390/diagnostics13182862_

Round 1

Reviewer 1 Report

INSTRUCTIVE AND WELL WRITTEN REVIEW ARTICLE !

EASY TO READ BUT, RECOMMEND COMPREHENSIVE REVISION WITH native English. 

Author Response

Dear Reviewer,

Thank you very much for your time and very useful suggestions for the revision of the Manuscript.

We enclose a corrected version of the paper, as well as a response to your suggestions and corrections.

With respect,

Dusan Popovic

Reviewer: EASY TO READ BUT, RECOMMEND COMPREHENSIVE REVISION WITH native English.

Answer: Corrected. The English language has been corrected.

Reviewer 2 Report

Dear authors,

This is an interesting narrative review. It is logically constructed and sets out the importance of artificial intelligence in some gastrointestinal diseases, underlying also that the human knowledge and experience cannot be totally eliminated. 

The available research is clearly presented and discussed, and the conclusion is supported by the evidence presented. All the bibliographic data are recent and very well structured in tables.

Return to the article: The abstract s a little large, please remake it! The abstract should be a total of about 200 words maximum, you have 243 now.  

I think your article is very important for the new generation of gastroenterologist and for newer concept of diagnostic and treatment.

Good luck!

Author Response

Dear Reviewer,

Thank you very much for your time and very useful suggestions for the revision of the Manuscript.

We enclose a corrected version of the paper, as well as a response to your suggestions and corrections.

With respect,

Dusan Popovic

Reviewer: The abstract s a little large, please remake it! The abstract should be a total of about 200 words maximum, you have 243 now. 

Answer: Corrected. The abstract is summarized in 200 words.

Reviewer 3 Report

Dear authors,

I have read your paper with interest. I find it well-written and welcomed in an era that I personally consider will be dominated by AI-driven technology.

I have already read back in 2022 the article of Francesco Renna et al, published in the same as your submission - ”Diagnosis” journal, which discussed the same topic as you have now. I think there are some concepts that can be worth discussing in the Introduction part of your paper (I disclose no interest in being cited, as I am not one of the authors of that paper).

The tables summarize well important studies in the AI field, yet you should proceed with caution when writing  ”the most important studies”. I think you can construct a section like”Material and Methods” where you can transform your review into a systematic one, searching according to the PRISMA guidelines the papers related to this topic, and after a careful selection discussing the most relevant or the ones that had the best impact in the academic world.

I feel your paper could be a good addition to the body of knowledge on this topic, and I am looking forward to seeing whether you feel you can transform it to become a systematic review and not only a ”fair point-of-view narrative review”. Good luck!

Dear authors,

I think the quality of the English language through the manuscript is fine and only minor spelling checks should be performed.

Author Response

Dear Reviewer,

Thank you very much for your time and very useful suggestions for the revision of the Manuscript.

We enclose a corrected version of the paper, as well as a response to your suggestions and corrections.

With respect,

Dusan Popovic

Reviewer #1: I have already read back in 2022 the article of Francesco Renna et al, published in the same as your submission - ”Diagnosis” journal, which discussed the same topic as you have now. I think there are some concepts that can be worth discussing in the Introduction part of your paper (I disclose no interest in being cited, as I am not one of the authors of that paper).

Answer #1: Corrected. Concepts from the mentioned manuscript are cited in the introduction.

Reviewer #2: The tables summarize well important studies in the AI field, yet you should proceed with caution when writing  ”the most important studies”.

Answer #2: Corrected. We fully agree with you that the wording "the most significant study" is too strong, and we have corrected it to "selected studies".

Reviewer #3: I think you can construct a section like”Material and Methods” where you can transform your review into a systematic one, searching according to the PRISMA guidelines the papers related to this topic, and after a careful selection discussing the most relevant or the ones that had the best impact in the academic world. I feel your paper could be a good addition to the body of knowledge on this topic, and I am looking forward to seeing whether you feel you can transform it to become a systematic review and not only a ”fair point-of-view narrative review”.

Answer #3: Thank you for the very justified and constructive suggestion. During the conceptualization of this manuscript, we had a dilemma whether it should be a review or a systematic review. We decided on a review because we thought it would be more readable and understandable in this form, especially since it is a complex and relatively new field of endoscopy. Also, we thought that this form would be a good starting point for future research in this area. Unfortunately, we think that at this stage we would not be able to adequately transform it into a systematic review. Sorry and thank you again.

Reviewer #4:I think the quality of the English language through the manuscript is fine and only minor spelling checks should be performed.

Answer #4: Corrected. The English language has been corrected.